# Research on Processing Technology of Multi-Layer Heterogeneous Material Composite Micron Cantilever Beam Structure

**DOI:** 10.3390/mi13020144

**Published:** 2022-01-18

**Authors:** Yingqi Shang, Hongquan Zhang, Zuofei Wu, Dongsa Chen, Shuangyu Wu

**Affiliations:** 1Electronic Science and Technology, College of Electronic Engineering, Heilongjiang University, Harbin 150001, China; 1202879@s.hlju.edu.cn; 2China Electronics Technology Group Corporation, The 49th Research Institute, Harbin 150001, China; 1202876@s.hlju.edu.cn (Z.W.); 1202877@s.hlju.edu.cn (D.C.); 02738@qqhru.edu.cn (S.W.)

**Keywords:** cantilever beam, porous silicon, sacrificial layer

## Abstract

In order to overcome the simplicity and instability of micron cantilever membrane structure, sacrificial layer technology and multi-layer heterogeneous material composite stacking technology were designated. Based on the research with respect to a multi-layer heterogeneous cantilever beam structure, multi-layer heterogeneous material composite, and sacrificial layer release craft, different characteristics of sacrificial layer material and film preparation craft were analyzed. According to these results, the suitable film preparation craft was generated to reduce the stress between materials and to improve the reliability and percentage of finished products. Our work put multi-layer material composite micro-cantilever beam structure into practice, and accelerated the manufacturing of a micro-acceleration sensor and vibration sensor in the future.

## 1. Introduction

With the rapid development of information technology, sensor technology has become the key to science and technology in the 21st century. It plays a decisive role in economic construction and technological development, and it is also an important indicator to evaluate the development level of a country [1]. MEMS (Micro Electro Mechanical System) technology is the inheritance and development of microelectronics technology. Due to the miniaturization of scale and the integration of dimension, it has become a new scientific field involving optics, medicine, electronic engineering, mechanical engineering, and other multidisciplinary cross-discipline [2]. MEMS devices and micromachining technology have the characteristics of miniaturization, microelectronics integration, and high precision [3,4,5,6,7,8,9,10,11]. Because of the influence of residual stress generated during the microstructure processing process of MEMS devices, the performance of sensors is degraded and the service environment of sensors is also limited. Therefore, it is proposed to use porous silicon as a sacrificial layer to realize the processing of silicon microstructures. Porous silicon is a unique “quantum sponge” or “branched” microstructured silicon-based material with a unique nano-silicon cluster framework. It is an important microfabricated material. The tower has a large surface area, good biocompatibility, and can emit light at room temperature. And the feature of adjustable refractive index has been widely used in the manufacture of miniature sensors and actuators. Porous silicon has a very large surface area due to its pore structure, and its surface activity is high. The pore surface adsorbs organic gas, inorganic gas, and water vapor. Therefore, it can be used as a sensitive material for gas and humidity sensors. Although porous silicon is filled with thick porous silicon, its structure is still single crystal. Using its single crystal characteristics, it can grow epitaxial silicon, oxide layers, and nitride layers of different materials on it. The thickness is large and can be accurately controlled, and the removal rate is high. It can be used as a sacrificial layer material in surface micromachining technology. The vigorous development of MEMS technology provides a strong impetus for the research of various microstructures, microsensors, and microactuators. Due to its low cost, light weight, low power consumption, small size, high sensitivity, and high response speed, the cantilever beam has become one of the common microstructures in MEMS. Micro cantilever beams are used in many fields, such as physical quantity detection, biochemical sensing, environmental detection, etc. There are various shapes of cantilever beams, among which the main ones are rectangular, triangular, T-shaped, U-shaped, tuning fork-shaped, bridge-shaped, and so on. Different shapes of microbeams are usually used in different fields, among which the rectangular shape is the most widely used.

Due to the thin film manufacturing process (deposition, evaporation, sputtering, etc.), the phenomenon of thermal mismatch, lattice mismatch, and residual stress will be generated. Residual stress will not only cause the deformation of the film, but also affect the performance and reliability of the device. Therefore, the influence of residual stress must be fully considered in the design, processing, and packaging process. Furthermore, the research on multilayer residual stress has received more attention [3]. As the residual stress in the suspended multilayer composite film is released after the sacrificial layer is removed, the microstructure is deformed. This deformation is the result of the combined effect of the residual stress in each layer of the film. Choosing a suitable thin film preparation process can effectively reduce the stress between heterogeneous materials. Selecting a reasonable sacrificial layer preparation process can complete the production of multi-layer material composite micron cantilever structure and lay the foundation for micro-acceleration sensors and vibration sensors.

## 2. Theoretical Model of Multi-Layer Heterogeneous Material Composite Cantilever Beam

Multilayer composite film is a common MEMS structure. For the suspended multilayer composite film structure, the residual stress in the film is released after removing the sacrificial layer. That results in the deformation of the microstructure. This deformation is the result of the combined effect of the residual stresses between layers of films. The characteristics of film deformation are more complicated due to the material properties of each layer being inconsistent. Testing techniques for thin film materials generally include substrate curvature testing method [12], microbeam rotation method [13], resonance frequency method [14], tympanic membrane method [15], nanoindentation method [16], and so on. This article uses the substrate curvature test method.

As the residual stress exists between the layers of the cantilever beam, it is easy to deform (compress or stretch) the cantilever beam, which eventually leads to the bending of the composite beam. We assume that they are all plain. Some layers are relatively elongated, while others are relatively shortened. At the interfaces between the elongation layer and the shortening layer, there is a layer that neither elongation nor shortening deformation occurs to, which is called the neutral layer of the beam or the neutral surface. The intersection of the neutral layer and the cross section of the beam becomes the neutral axis. Set the *x*-axis as the neutral axis of the composite layer, along the width direction as the *y*-axis, which is perpendicular to the beam plane, and along the thickness direction as the *z*-axis to establish a rectangular coordinate system. The bottom layer is the first layer, and the top layer is the nth layer. The width and thickness of each layer is w_1_, w_2_, wn and h_1_, h_2_, h_n_, and the Young’s modulus; the Poisson’s ratio and residual stress of each layer is E_i_, v_i_, and σ_i_ (subscript i represents the layer number). The distance between the *x*-axis and the lower surface of the cantilever beam is z_c_. The structure is shown as in Figure 1. 

Static balance in cross section:(1)$$\int_{-z_{c}}^{h_{1}-z_{c}}E_{1}w_{1}\mathrm{zdz}+\int_{h_{1}-z_{c}}^{h_{1}+h_{2}-z_{c}}E_{2}w_{2}\mathrm{zdz}+\ldots +\int_{h_{1}+h_{2}+\ldots +h_{n-1}-z_{c}}^{h_{1}+h_{2}+\ldots +h_{n}-z_{c}}E_{n}w_{n}\mathrm{zdz}=0$$
Neutral axis position:(2)$$z_{c}=\frac{1}{2}\frac{\sum_{i=1}^{n}E_{i}w_{i}h_{i}\left[\left(\sum_{j=1}^{i}2h_{j}\right)-{\;h}_{i}\right]}{\sum_{i=1}^{n}E_{i}w_{i}h_{i}}$$
Equivalent bending stiffness of composite beam:
(3)$${\mathrm{EI}}_{\mathrm{eff}}=\int_{-z_{c}}^{h_{1}-{\;z}_{c}}E_{1}w_{1}z^{2}\mathrm{dz}+\ldots +\int_{h_{1}+h_{2}+\ldots +h_{n\;-1}-z_{c}}^{h_{1}+h_{2}+\ldots +h_{n}-{\;z}_{c}}E_{n}w_{n}z^{2}\mathrm{dz}=\overset{n}{\underset{i=1}{\sum }}E_{i}w_{i}h_{i}\left\{\frac{h_{i}^{2}}{12}+{\left[\left(\overset{i}{\underset{j=1}{\sum }}h_{j}\right)-\frac{h_{i}}{2}-z_{c}\right]}^{2}\right\}$$
Bending moment of composite beam:(4)$$\begin{array}{cc} M=\int_{-z_{c}}^{h_{1}-{\;z}_{c}}\sigma_{1}w_{1}\mathrm{zdz} & +\int_{h_{1}-{\;z}_{c}}^{h_{1}+h_{2}-{\;z}_{c}}\sigma_{2}w_{2}\mathrm{zdz}+\ldots +\int_{h_{1}+h_{2}+\ldots +h_{n-1}-z_{c}}^{h_{1}+h_{2}+\ldots +h_{n}-{\;z}_{c}}\sigma_{n}w_{n}\mathrm{zdz}\\ & =\frac{1}{2}\sum_{i=1}^{n}\sigma_{i}w_{i}h_{i}\left[\left(\sum_{j=1}^{i}2h_{j}\right)-{\;h}_{i}-2z_{c}\right] \end{array}$$
$$M=\int_{-z_{c}}^{h_{1}-{\;z}_{c}}\sigma_{1}w_{1}\mathrm{zdz}+\int_{h_{1}-{\;z}_{c}}^{h_{1}+h_{2}-{\;z}_{c}}\sigma_{2}w_{2}\mathrm{zdz}+\ldots +\int_{h_{1}+h_{2}+\ldots +h_{n\;-1}-z_{c}}^{h_{1}+h_{2}+\ldots +h_{n}-{\;z}_{c}}\sigma_{n}w_{n}\mathrm{zdz}$$
Radius of curvature:(5)$$R=\left|\frac{{\mathrm{EI}}_{\mathrm{eff}}}{M}\right|$$

## 3. Preparation of Multi-Layer Heterogeneous Material Composite Cantilever Beam

### 3.1. Experiment Procedure

The preparation of the well-released sacrificial layer with good support effect is the key to realizing the cantilever beam structure. Generally, photoresist, SiO_2_, or porous silicon are used as the sacrificial layer to realize the release of the suspended structure. The production of porous silicon is simpler than other methods [17,18,19]. Due to its controllable porosity and thickness and high removal selectivity, porous silicon is used as the sacrificial layer in this article. Porous silicon is prepared in a specific area by electrochemical etching technology [20,21,22]. The porous silicon sacrificial layer with good performance could be prepared by controlling the current density and solution concentration [23,24].

A P-type double polished silicon wafer with <100> crystal orientation, a thickness of 400 μm, and a resistivity of 0.01~0.02 Ω·cm^−1^ was used in this article. The porous silicon layer was prepared by a dual-channel electrochemical etching method. The size of the cantilever beam was 1400 μm × 300 μm × 2 μm (length × width × thickness). The technological process is shown as in Figure 2.

In the above process, the preparation of the sacrificial layer is the key to the release of the cantilever beam. In order to prevent the cantilever beam from being damaged when it is pressed down, the thickness of the porous silicon layer is controlled within 20 μm. The electrolyte solution used in this paper is a mixture of 15%HF and anhydrous ethanol after our many experiments. And the distance between the electrode and the silicon wafer is fixed. When the current density is 50 mA/cm^2^, the film made on the surface of the porous silicon layer is smooth and flat, and the porosity of the prepared sacrificial layer is controllable. The removal rate is moderate when the cantilever is released, which can effectively reduce the probability of damage to the cantilever.

### 3.2. Experiment Results

Experiments were carried out using the mixed solution of etchant HF and C2H5OH, the etching time was 10 min, and the current density was 10 mA/cm^2^ as the preparation method of porous silicon. Figure 3 shows the SEM photograph of the microstructure of the porous silicon by our method.

First, measure the radius of the curvature of the film according to the design size. This parameter can directly reflect the height of the film stress. The film thickness is shown in Table 1. The test result is shown in Figure 4.

The text continues here (Figure 4 and Table 2).

From Figure 4, it is found that the stress of the SiO_2_ film and the SiN_x_ film are significantly greater than the stress of the metal film and the sensitive film. Therefore, in the design process, the stress of the SiO_2_/SiN_x_ film is mainly improved to reduce the bending of the cantilever beam after the release process. The length and width dimensions of the cantilever are fixed, and the SiO_2_/SiN_x_ different thickness matching design is carried out. The design process parameters in the experiment are shown in Table 2.

The cantilever beam is released according to the process flow shown in Figure 2. The sample is shown in the Figure 5.

The cantilever beam is released according to the process flow shown in Figure 2, the sample is shown in the Figure 6.

Fixing the size of the cantilever beam, it can be seen that the deformation of the composite film decreases with the decrease in the thickness of SiO_2_, and decreases with the increases in the thickness of the SiN_x_ from the Figure 4. This is because the SiO_2_ film presents compressive stress during the production process, and the SiNx film presents tensile stress during the production process. The stress produced by SiO_2_ of the same thickness is greater than the stress produced by SiN_x_. After the sacrificial layer is removed, the stress is released and the cantilever beam is deformed. The thickness ratio of SiO_2_/SiN_x_ changes the position of the neutral axis of the composite film, changes the radius of curvature, and improves the deformation of the cantilever caused by the internal residual stress after the cantilever is released.

Based on the above experimental results, a piezoelectric cantilever beam was prepared. The metal bottom electrode, the piezoelectric film, and the metal top electrode could be grown on the SiO_2_/SiN_x_ film substrate to realize the cantilever beam structure. The prepared multi-layer heterogeneous material composite cantilever beam structure was released well, as shown in the Figure 7.

## 4. Conclusions

By studying the influence of etching solution, current density, etching time, and doping method on the preparation of porous silicon, we tested the influence of different process parameters. The relationship between process parameters and porous silicon microstructure, porous silicon thickness, and porosity was obtained. Optimizing process parameters and choosing different doping forms can control the pore size and thickness of porous silicon. Utilizing the characteristics of porous silicon and using porous silicon as a sacrificial layer, a silicon microcantilever structure is realized. The influence of different film thickness on the deformation of the cantilever beam after release is studied. Conclusion: choosing a suitable film thickness can realize the preparation of low-stress cantilever beam structure, and effectively improve the bending deformation of a multi-layer heterogeneous material composite cantilever beam. It can be used in acceleration, vibration sensors, and other fields.

## Figures and Tables

**Figure 1 micromachines-13-00144-f001:**
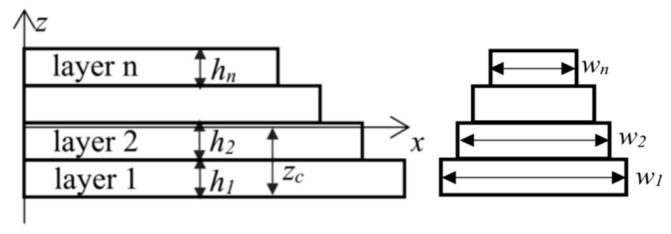
Theoretical model of a multi-layer composite cantilever beam.

**Figure 2 micromachines-13-00144-f002:**
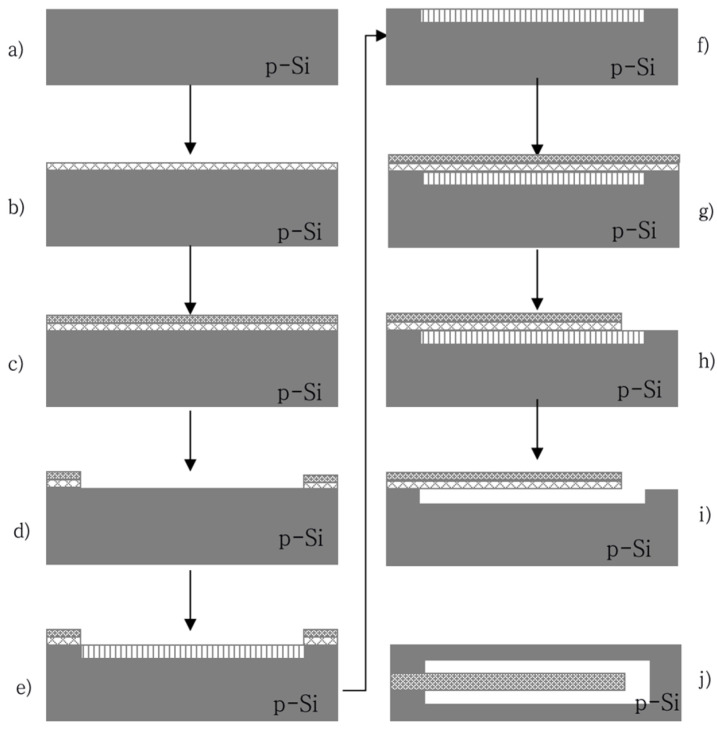
Schematic diagram of the cantilever beam process. (**a**) Cleaning the substrate. The standard general cleaning in the MEMS process is used to clean the surface of the substrate. (**b**) Manufacture of SiO_2_ film using thermal oxidation technology. It is used for the patterned mask layer of porous silicon, the thickness is 120 nm. (**c**) Growing Si_3_N_4_ by LPCVD(Low Pressure Chemical Vapor Deposition). It is used for the patterned mask layer of porous silicon, the thickness is 180 nm. (**d**) Photoetch the porous silicon preparation area to protect the front side and remove the dielectric layer on the back side. Make the substrate have certain conductivity and improve the uniformity of porous silicon preparation. (**e**) Electrochemical preparation of porous silicon [20,21,22]. The mixed solution of 15% HF and absolute ethanol is used as the electrolyte. The current density is set to 50 mA/cm^2^, the film formed on the surface of the porous silicon layer is smooth and flat, and the porosity of the prepared sacrificial layer is controllable. And when the cantilever is released, the removal rate is moderate, which can effectively reduce the probability of damage to the cantilever. Molybdenum is used as a double-slot electrochemical corrosion electrode, the distance between the electrode and the silicon wafer is fixed, and the electrification time is 30 min. The thickness of porous silicon can reach 20 μm. (**f**) Removing the SiO_2_/Si_3_N_4_. After electrochemical etching, the SiO_2_/Si_3_N_4_ film on the surface is removed, and the substrate is cleaned at the same time, in preparation for the subsequent growth of the multi-layer heterogeneous cantilever beam film. (**g**) Multi-layer heterogeneous film is composed of SiO_2_/SiN_x_, metal electrodes and piezoelectric sensitive film between metal layers. Use PECVD (Plasma Enhanced Chemical Vapor Deposition) to make SiO_2_/SiN_x_ film, and magnetron sputtering to make metal film and sensitive film. The schematic diagram is shown in Figure 2g. (**h**) Use photolithography, wet etching and RIE (reactive ion etching) technology to fabricate cantilever structures. (**i**) Use wet etching technology to remove porous silicon to achieve cantilever beam release. Using a KOH solution with a concentration of 0.5% can effectively remove porous silicon without causing damage to the cantilever film. Au is used as the metal electrode material. Put the substrate in the KOH solution and take it out after 30 min. (**j**) The top view of the cantilever beam structure.

**Figure 3 micromachines-13-00144-f003:**
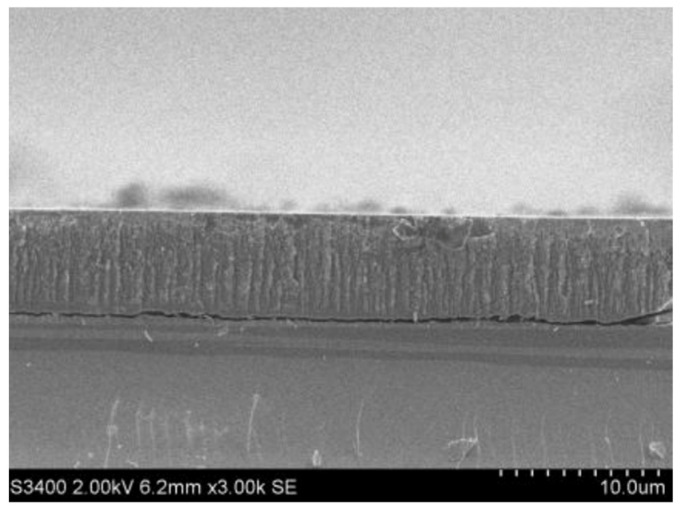
Preparation of porous silicon microstructure.

**Figure 4 micromachines-13-00144-f004:**
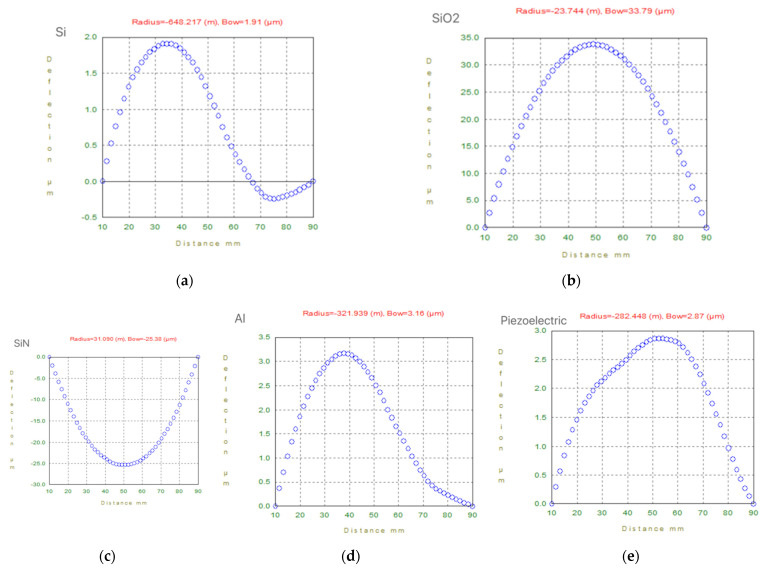
The test results of the radius of curvature of the film. (**a**) Si; (**b**) SiO_2_; (**c**) SiN; (**d**) Al; (**e**) Piezoelectric.

**Figure 5 micromachines-13-00144-f005:**
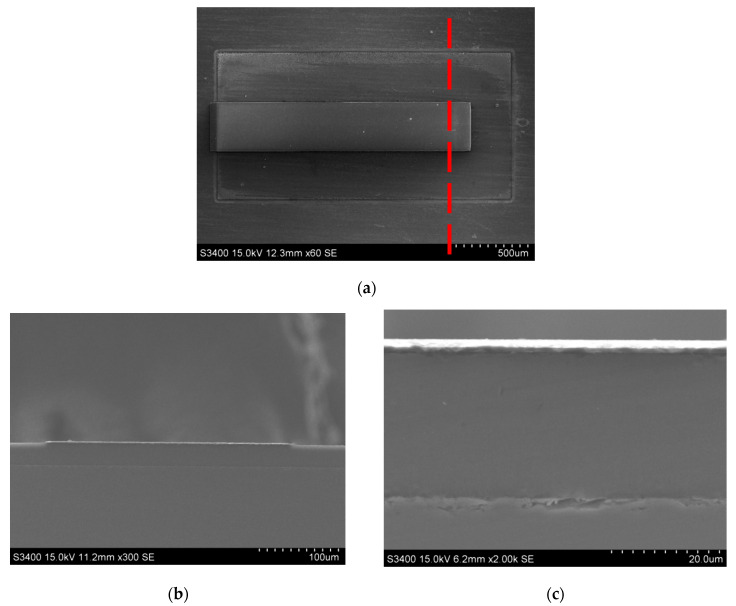
Preparation of porous silicon microstructure. (**a**) Top view. (**b**) Sectional view (overall). (**c**) Sectional view (partial).

**Figure 6 micromachines-13-00144-f006:**
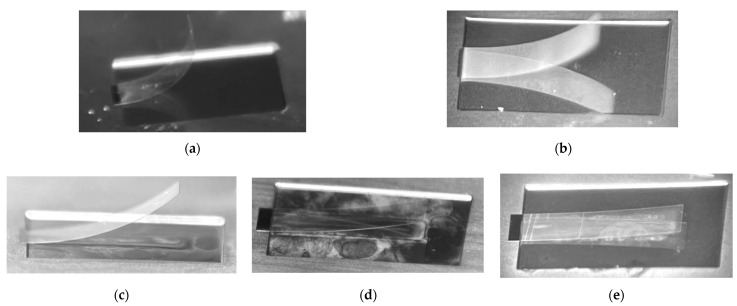
The Physical image of a cantilever beam with different SiO_2_/SiN_x_ thickness. (**a**) 1# SiO_2_/SiN_x_: 1.25/0.75 (μm); (**b**) 2# SiO_2_/SiN_x_: 1/1 (μm); (**c**) 3# SiO_2_/SiN_x_: 0.75/1.25 (μm); (**d**) 4# SiO_2_/SiN_x_: 0.5/1.5 (μm); (**e**) 5# SiO_2_/SiN_x_: 0.25/1.75 (μm).

**Figure 7 micromachines-13-00144-f007:**
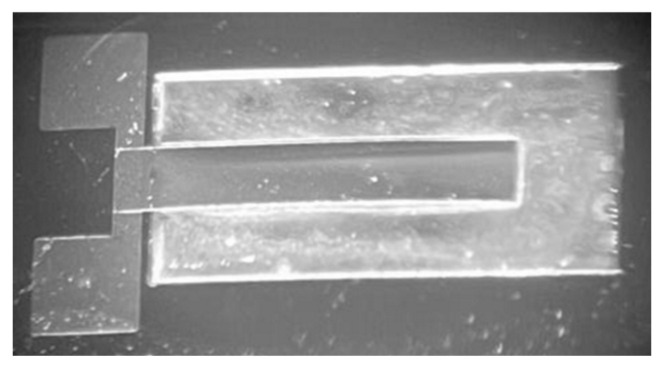
Real photo of multilayer composite cantilever with piezoelectric sensitive film and metal film.

**Table 1 micromachines-13-00144-t001:** Film thickness.

Material	Thickness (μm)
Si	400
SiO_2_	1
SiN_x_	1
Al	0.2
Piezoelectric	0.2

**Table 2 micromachines-13-00144-t002:** This Process experimental parameters.

Sample	Process Conditions	
	SiO_2_ Film Thickness (μm)	SiN_x_ Film Thickness (μm)
1#	1.25	0.75
2#	1	1
3#	0.75	1.25
4#	0.5	1.5
5#	0.25	1.75

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
