# Peer review of "Research on Processing Technology of Multi-Layer Heterogeneous Material Composite Micron Cantilever Beam Structure"

_micromachines, 2022, doi:10.3390/mi13020144_

Round 1
Reviewer 1 Report
The paper reports on the fabrication of multilayer cantilever structure using porous silicon as a sacrificial layer. However, the paper needs a significant improvement both in terms of experimental data and presentation of the results.
I'm happy to provide some technical points to be addressed in the revision process:
1) English of the paper must be improved.
2) I think it would be useful for the readers to have a broader pictures of the potential of the electrochemical etching for silicon micro machining and MEMS. Please, find a few examples of papers about the electrochemical micromachining of silicon:
- Controlled Microfabrication of High‐Aspect‐Ratio Structures in Silicon at the Highest Etching Rates: The Role of H2O2 in the Anodic Dissolution of Silicon in Acidic Electrolytes, Advanced Functional Materials 27 (6), 1604310 (2017)
- Electrochemical micromachining as an enabling technology for advanced silicon microstructuring, Advanced Functional Materials, 22 (6), 1222-1228 (2012)
- Electrochemical etching in HF solution for silicon micromachining, Sens. Actuators A, 102, 195-201 (2002).
3) When discussing porous silicon integration on silicon wafer, please, consider discussing state-of-art works reporting the integration of porous silicon in commercial microelectronic processes:
- Peripheral Nanostructured Porous Silicon Boosts Static and Dynamic Performance of Integrated Electronic Devices, Advanced Electronic Materials 6 (12), 2000615 (2020)
-CMOS-compatible fabrication of porous silicon gas sensors and their readout electronics on the same chip, Phys. Stat. Sol.(a) 204 (5), 1423-1428 (2007)
- An integrated CMOS sensing chip for NO2 detection, Sensors and Actuators B: Chemical 134 (2), 585-590 (2008)
4) Details on the experimental fabrication and characterisation of the cantilever structure are not fully provided. I would suggest to add experimental parameters on the the different steps of the fabrication process and data on the experimental characterization.
5) Please, provide SEM picture of the cantilever structure before removal of the PSi layer, both top view and crosse section.
6) Organization of the paper and presentation and discussion of the results must be significantly improved.
Author Response
Dear reviewer,
The revised manuscript has been uploaded as an attachment, please check.

Reviewer 2 Report
This paper draws some options to improve the bending behavior of composite cantilever beam with multi-layer heterogeneous composite material. It could give some interesting points in sensor production for acceleration or vibration detection problems.
- The first section is numbered by 0.
- The introduction is relatively short, and there are a little references to the related works. It needs some extension.
- In the second section (numbered by 1) there are no references at all. Theoretical background needs to clarify.
- In Figure 1. font sizes should be harmonized.
- Eq. (1) is not an equation or something is missing.
- In section 3 (numbered by 2) there are many physical quantity with units. But between the numbers and the units could be some extra space to help readability.
- On page 3 line 94: what is the meaning of P<100>?
- Some references or description about LPCVD or PECVD techniques could help the reader to understand.
- The "Result" section title should keep together the section.
- There are some Fig. (Tab.) and Figure (Table) referencing style as well. It could be written in uniformly.
- The paper contains several typo errors which need correction: extra spaces, missing spaces, extra commas, missing sentence terminator, miss-spelling words.
Author Response

(The authors gave the same response as above.)

Round 2
Reviewer 1 Report
Authors did some of the suggested changes, but in my opinion the manuscript still need to be thoroughly reworked.
The main aim of a scientific manuscript are to report the experimental results as well as to reports the details of the method, so that any other research can, in principle, reproduce exactly the same process.
In this case the authors just did some minor modifications by adding the suggested references (please, adjust the author name and citation style in the references added) and adding some detail of the process. However, most the process parameters are still missing, such as details of steps a) through j). Also, the SEM pictures added do not clearly highlight the porous silicon layer under the cantilever structures.
In my opinion the manuscript cannot be published in its present form and need to be thoroughly reworked and organized, as per my comments in the first review round.
Author Response
Thank you for your comment. The attachment has been uploaded and modified.

Round 3
Reviewer 1 Report
In the following references author's name is misspelled and must be corrected as in follows:
38: A. Paghi, L. M. Strambini, F. F. Toia, M. Sambi, M. Marchesi, R. Depetro, M. Morelli, G. Barillaro, Peripheral Nanostructured Porous Silicon Boosts Static and Dynamic Performance Of Integrated Electronic Devices, Advanced Electronic Materials 6, 2000615 (2020).
41: C. Cozzi, G. Polito, K. Kolasinski, G. Barillaro, Controlled Microfabrication of High-Aspect-Ratio Structures in Silicon at the Highest Etching Rates: The Role of H2O2 in the Anodic Dissolution of Silicon in Acidic Electrolytes, Advanced Functional Materials, 7, 1604310(2017).
42: M. Bassu, S. Surdo, L. M. Strambini, G. Barillaro, Electrochemical micromachining as an enabling technology for advanced silicon microstructuring, Advanced Functional Materials, 22 (6), 1222-1228 (2012).
Author Response
Please see the attachment.

This manuscript is a resubmission of an earlier submission. The following is a list of the peer review reports and author responses from that submission.